# LDHA-mediated ROS generation in chondrocytes is a potential therapeutic target for osteoarthritis

Manoj Arra [1], Gaurav Swarnkar[1], Ke Ke[1], Jesse E. Otero[2], Jun Ying[1], Xin Duan, Takashi Maruyama [3,6], Muhammad Farooq Rai [1], Regis J. O'Keefe[1], Gabriel Mbalaviele[4], Jie Shen[1] & Yousef Abu-Amer [1,5 ✉]

The contribution of inflammation to the chronic joint disease osteoarthritis (OA) is unclear, and this lack of clarity is detrimental to efforts to identify therapeutic targets. Here we show that chondrocytes under inflammatory conditions undergo a metabolic shift that is regulated by NF-κB activation, leading to reprogramming of cell metabolism towards glycolysis and lactate dehydrogenase A (LDHA). Inflammation and metabolism can reciprocally modulate each other to regulate cartilage degradation. LDHA binds to NADH and promotes reactive oxygen species (ROS) to induce catabolic changes through stabilization of IκB-ζ, a critical pro-inflammatory mediator in chondrocytes. IκB-ζ is regulated bi-modally at the stages of transcription and protein degradation. Overall, this work highlights the function of NF-κB activity in the OA joint as well as a ROS promoting function for LDHA and identifies LDHA as a potential therapeutic target for OA treatment.

[1] Department of Orthopaedic Surgery and Cell Biology & Physiology, Washington University School of Medicine, St. Louis, MO 63110, USA. [2] OrthoCarolina Hip and Knee Center, Charlotte, NC 28207, USA. [3] Department of Immunology, Akita University School of Medicine, Akita, Japan. [4] Bone and Mineral Division, Department of Medicine, Washington University School of Medicine, St. Louis, MO 63110, USA. [5] Shriners Hospital for Children, St. Louis, MO 63110, USA. [6]Present address: Mucosal Immunology Section, National Institute of Dental and Craniofacial Research, National Institutes of Health, Bethesda, MD 20892, USA. ✉email: abuamery@wustl.edu

Osteoarthritis (OA), a degenerative disease of the joint, is the most prevalent musculoskeletal disorder affecting over 50 million in the United States and 200 cases per 100,000 individuals globally, and the leading cause of disability[1–4]. However, despite its prevalence and severity, there are no curative or effective disease-modifying treatments for OA owing to poor understanding of disease pathogenesis. In healthy joints, mechanical loading and biochemical cues maintain articular homeostasis via a balance between chondrocyte synthesis of anabolic factors like type II collagen and aggrecan, and catabolic factors like matrix metalloprotease 13 (MMP13) and A Disintegrin and Metalloproteinase with Thrombospondin motifs (ADAMTS)[5,6]. Since extracellular anabolic matrix molecules are extremely stable, reduction of catabolic enzymes may be sufficient to inhibit OA disease progression, especially at early stages of disease[7]. Several studies have shown that both the synovial fluid and serum of OA patients have elevated levels of pro-inflammatory mediators such as interleukin (IL)-1β[8–10] that drive the production of catabolic degradative enzymes in chondrocytes[11,12]. Mechanistically, many inflammatory stimuli converge upon the NF-κB pathway, considered the gatekeeper of the inflammatory response in cells[13,14]. Thus, activation of NF-κB is proposed to play a key role in OA progression by promoting the expression of cartilage degrading genes and chondrocyte dysfunction, though the mechanisms by which pathologic activation of NF-κB induces joint catabolism in OA are obscure.

Energy metabolism is an extremely important mediator of cellular function often altered during disease states, especially under chronic inflammatory conditions. It has been shown that chondrocytes in OA cartilage undergo metabolic changes though the role of these changes in disease pathology is unclear[15–19]. Furthermore, recent work has indicated that chondrocytes can undergo metabolic changes in response to various stimuli[23,25,26]. Chondrocytes in vivo are proposed to rely heavily upon glycolysis, with lower energy production coming from oxidative phosphorylation (OxPhos) due to the relatively hypoxic environment they reside within[18]. Glycolysis, which is oxygen-independent, generates ATP quickly but inefficiently, while TCA cycle and electron transport chain (ETC) are extremely efficient at energy production through OxPhos when oxygen is available, though most cells during physiological conditions take advantage of a combination of glycolysis and OxPhos[20]. However, stressed cells, such as immune and cancer cells, paradoxically utilize glycolysis and fermentation to generate ATP even in the presence of oxygen, i.e., aerobic glycolysis[21–23]. This altered cellular metabolism is not only important for energy homeostasis, but likely critical for altered cellular function and signaling[24–26]. We propose that inflammation-driven metabolic reprogramming is important for sustaining and driving inflammatory response pathways in chondrocytes, a paradigm for understanding their co-regulation and relationship to disease.

The inflammatory response is also driven by myriad factors[14,27]. Reactive oxygen species (ROS) are one such player considered as potential OA disease culprits due to their ability to modify proteins and lipids, damage DNA and other adverse effects in cells[28–31]. While ROS play important signaling functions in physiological states, elevated ROS present significant pathologic risk as mediators of disease progression in OA[32]. Furthermore, the production and elimination of ROS is closely tied to metabolic pathways, with enzymes and substrates playing dual roles in both ROS modulation and metabolic processing[33,34].

Thus, we proposed that inflammation-mediated metabolic shifts may underlie joint degradation in OA by promoting the production of inflammatory and catabolic proteins through the modulation of ROS in chondrocytes. We display that NF-κB activation in OA drives metabolic reprogramming of chondrocytes towards aerobic glycoylsis. We then show that this metabolic reprogramming causes increased oxidative stress in the cell in an lactate dehydrogenase A (LDHA)-mediated manner. Finally, we demonstrate that metabolism-mediated oxidative stress promotes catabolic changes via the expression of IκB-ζ protein stabilization. Our findings highlight a relationship between these various players and offer therapeutic modalities for the treatment of OA.

## Results

**Inflammatory stimulation induces metabolic changes**. We initially sought to determine the effect of inflammatory cytokines on chondrocytes in order to identify downstream therapeutic targets. Treatment of primary murine chondrocytes with IL-1β, a pro-inflammatory cytokine implicated in OA[10], caused acidification of media (Supplementary Fig. 1A), suggesting a shift in cell metabolism akin to increased secretion of lactic acid via lactate dehydrogenase (LDH)-mediated fermentation[35]. Primary sternal chondrocytes were utilized immediately upon isolation without any expansion to avoid de-differentiation and altered metabolism. Next, mRNA sequencing showed that IL-1β caused a significant increase in the expression of catabolic and inflammatory genes such as MMPs and Interleukins, while decreasing the expression of chondrocyte anabolic factors such as aggrecan (Fig. 1a (arrows); Supplementary Data 1), among many other differentially regulated genes. Interestingly, when inflammatory and immune pathways were filtered out, RNA-seq analysis revealed a significant difference in several metabolic pathways in the KEGG database[36]. Gene set analysis indicated a decrease in OxPhos and other metabolic pathways (Table 1), that are important for glucose metabolism. We further detected a significant increase in expression of genes involved in glycolysis and fermentation such as Hexokinase-2 and LDHA with a decrease or no change in TCA cycle and ETC genes (Fig. 1b and Supplementary Data 1). Overall, these results suggest a shift in glucose metabolism of chondrocytes towards aerobic glycolysis.

We confirmed these findings by detecting elevated levels of extracellular lactic acid in supernatant of IL-1β-treated cell cultures (Supplementary Fig. 1B). Further, consistent with RNA-seq data, using Seahorse assay, we detected increased glycolysis and decreased OxPhos in response to IL-1β as measured by cell respiration (Fig. 1c–g). In addition, these cells appeared to have slightly altered, though insignificant, mitochondrial biogenesis upon IL-1β treatment, potentially indicating a compensatory response due to altered metabolic activity (Supplementary Fig. 1c). These cells also displayed lower intracellular ATP levels with inflammatory stimulation (Supplementary Fig. 1D). In accordance with these metabolic and RNA-seq data, GLUT1 and LDHA, the key enzyme involved in fermentation, were increased upon inflammatory stimulation (Fig. 1h–i). In addition, chondrocytes subject to inflammation had increased expression and activity of G6PD2, the rate limiting enzyme involved in the oxidative phase of pentose phosphate pathway (PPP) and modulation of NADPH levels (Fig. 1j, k and Supplementary Fig. 1E) with no significant change in expression of TCA cycle enzymes or non-oxidative phase of PPP (Supplementary Fig. 1F–J).

We then examined if the metabolic shift was mediated by NF-κB activation by IL-1β. Inhibition or deletion of IKK2, the classical NF-κB kinase, was able to reverse the metabolic shifts observed with IL-1β stimulation of chondrocytes (Fig. 1c–j, l; Supplementary Fig. 1B–D, and Supplementary Fig. 1F–J). Furthermore, expression of constitutively active IKK2 (IKK2ca)[51,52] in chondrocytes led to an increase in the expression of genes involved in

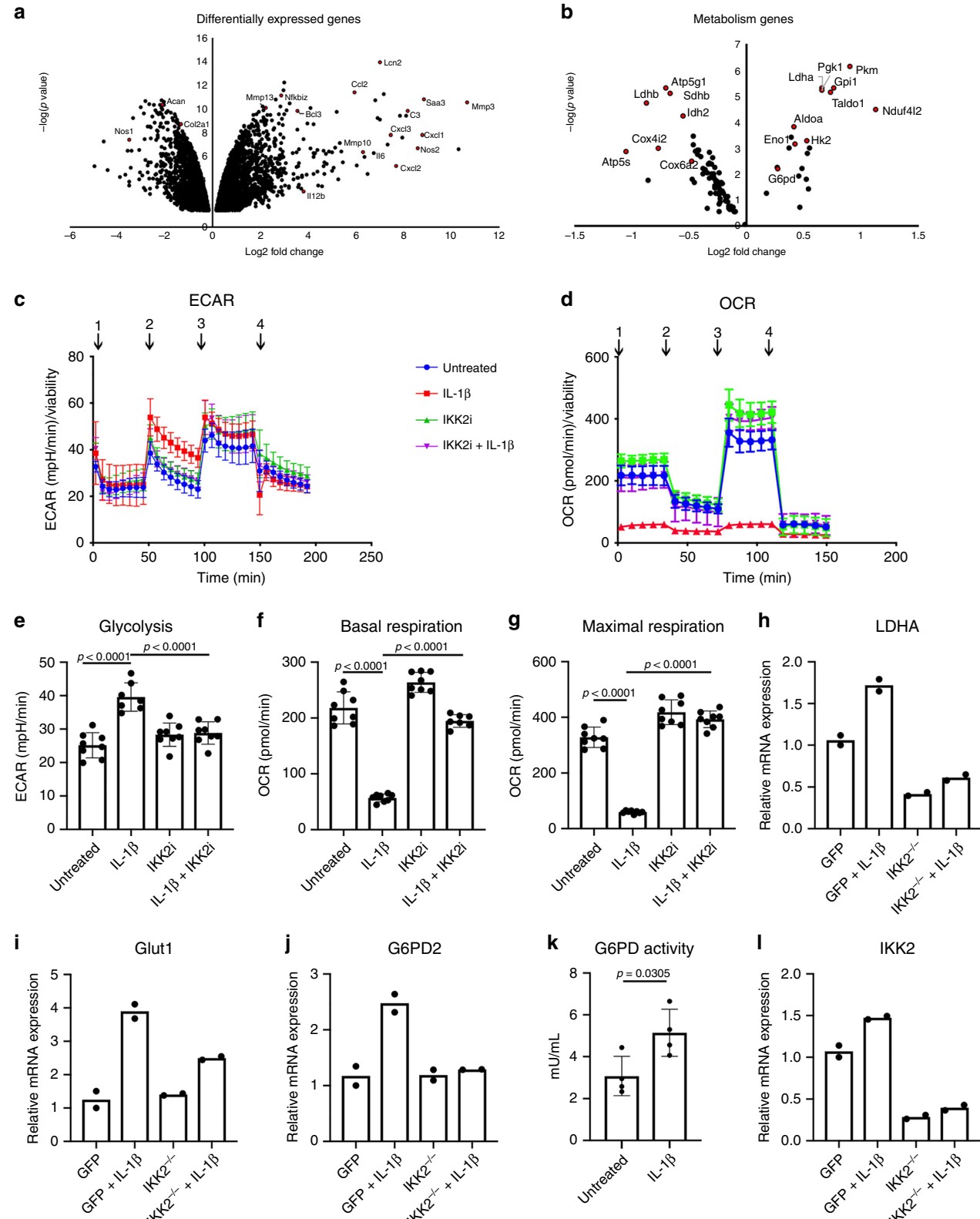

glycolysis independent of cytokine stimulation, confirming direct NF-κB-mediated effects (Supplementary Fig. 1K–M).

Finally, to determine the temporal regulation of metabolic changes in response to inflammation, a time course of IL-1β treatment of chondrocytes displayed that changes in metabolic enzyme gene expression occurred within 6 h of treatment, indicating that the shift in metabolism may occur soon after stimulation (Supplementary Fig. 1N–U). Furthermore, measurement of ECAR at these timepoints indicated that glycolytic activity was elevated at similar timepoints (Supplementary Fig. 1V–W). These results indicate an overall metabolic reprogramming of cellular glucose metabolic processes in response to inflammation.

**Fig. 1 Inflammation promotes aerobic glycolysis in chondrocytes.** RNA sequencing was performed on primary chondrocytes treated with IL-1β for 24 h and compared to untreated chondrocytes. Volcano plots were generated representing catabolic and anabolic genes (**a**) and metabolic genes (**b**). (**c**, **d**) Primary sternal chondrocytes were treated with IL-1β (10 ng/mL) in the presence or absence of IKK2i (10 μM) for 72 h prior to performing seahorse assay. All values were normalized to cell viability of treatments relative to untreated cells as measured by MTT assay. **c** ECAR measurement in glycolysis stress test (Injection 1: No treatment, Injection 2: Glucose, Injection 3: Oligomycin, Injection 4: 2-DG) or **d** OCR measurement in MitoStress test (Injection 1: No treatment, Injection 2: Oligomycin, Injection 3: FCCP, Injection 4: Antimycin A/Rotenone) was performed on Seahorse Instrument. Measurements were performed every 5 min with per timepoint for each condition. Graphs shown in **c**, **d** are mean ± S.D. for 8 wells, representative of one of three independent experiments. (Untreated = Blue, IL-1β = Red, IKK2i = Green, IKK2i + IL-1β = Purple). **e** Quantification of glycolysis from one timepoint in Glycolysis Stress Test or **f**–**g** quantification of basal respiration and maximal respiration for one timepoint each from MitoStress test (mean ± S.D. for $n = 8$ wells). **e**–**g** One-way ANOVA was performed followed by Tukey's Multiple Comparison's test, with p-values indicated in figure. **h**–**j** IKK2$^{f/f}$ chondrocytes were infected with adenoviral-GFP, or adenoviral-cre (labeled IKK2$^{-/-}$), then treated with IL-1β (10 ng/mL) for 24 h. qPCR as performed for LDHA, GLUT1, and G6PD2. Data shown is representative of one experiment out of three, with bars representing mean of duplicates. **k** G6PD activity was measured in primary chondrocytes treated with IL-1β (10 ng/mL) for 24 h compared to untreated cells. Bars are mean ± S.D. for $n = 4$ biological replicates. Two-tailed unpaired Student's t-test was utilized for statistical analysis. **l** Deletion of IKK2 under the conditions described for panels **h**–**j** was confirmed by performing qPCR for IKK2, with bars representing mean of duplicates.

**Table 1 Top KEGG metabolic pathways significantly differing with IL-1β treatment.**

| KEGG pathway | Mean log2 fold change | p-value |
|---|---|---|
| Oxidative phosphorylation | −2.9581 | 0.00178 |
| Butanoate metabolism | −2.2511 | 0.01604 |
| Valine, leucine, and isoleucine degradation | −2.1551 | 0.01702 |
| Glutathione metabolism | 1.92693 | 0.02858 |
| Citrate cycle | −1.7425 | 0.04515 |
| Fatty acid metabolism | −1.7062 | 0.04578 |
| Propanoate metabolism | −2.0355 | 0.02402 |

**NF-κB activation mediates injury-induced OA in mouse knee.** We surmised that in human OA patients, initial insults to the joint promote low-grade inflammation and activate NF-κB pathway to initiate disease progression, which is then sustained chronically. We first confirmed activation of NF-κB in the MLI model by performing surgery in 12-week-old transgenic mice expressing luciferase reporter under the control of the NF-κB promoter. In vivo bioluminescence imaging (BLI) showed that NF-κB activity peaked around 1 week post surgery in the joint, and was significantly higher in MLI compared to sham-operated joints (Fig. 2a, b). NF-κB activation declined progressively in both groups with time, but remained higher in MLI-operated compared to sham-operated joints, suggesting a chronically activated state within the joint space. Micro-computed tomography (μCT) and radiographic analyses showed that mice subjected to MLI developed subchondral bone sclerosis and decreased joint spacing 8 weeks after surgery (Fig. 2c, d; white arrows). These structural changes were associated with loss of proteoglycans and fibrillation of articular cartilage (Fig. 2e, green arrows), which mimic the changes seen in human OA patients[37]. Furthermore, immuno-histochemistry performed in joint sections showed increased p-IKK2 expression in articular cartilage (Fig. 2f; arrows). Buttressing the findings that NF-κB is activated in articular cartilage during OA development (Fig. 2f), we show that expression of NF-κB-induced catabolic enzymes is increased in cartilage of MLI joints (Supplementary Fig. 2A–D).

We scrutinized the direct role of NF-κB activation in OA pathogenesis via expression of IKK2ca. We and others have reported previously that IKK2ca mimics a chronically activated inflammatory state by activating NF-κB transcriptional machinery in the absence of exogenous inflammatory stimuli[38,39]. We confirmed that treatment of primary chondrocytes with IL-1β or exogenous expression of IKK2ca in these cells both activated NF-κB signaling (Fig. 3a). Expression of IKK2ca increased gene expression of inflammatory cytokines such as IL-6 and catabolic enzymes such as MMP13 and ADAMTS4 and pro-inflammatory factors such as MCP-1 (Fig. 3b–f). IL-6 and MMP13 are highly implicated in joint breakdown[40–43] and will be used as surrogates of inflammatory and catabolic gene expression throughout the study.

Based on the in vivo finding that MLI chronically activates NF-κB in articular cartilage and in vitro data displaying that NF-κB drives catabolic genes involved in OA, we expressed adenoviral IKK2ca in the joint and used LacZ expression as control. We observed that IKK2ca expression in the joint caused a severe joint degradation that resemble joints of end-stage rheumatoid arthritis, with significant inflammatory infiltrate and complete destruction of the femoral and tibial heads (Fig. 3g, arrows). However, this does not resemble the pathology seen in OA, suggesting that the inflammatory response in OA is more cell specific and occurs in a chronic, low-grade manner. Thus, in order to create a more specific model, we genetically expressed IKK2ca in articular chondrocytes in adult mice using the tamoxifen-inducible aggrecan (Acan1)-CreERT2. Histological analysis of joints by safranin-O staining revealed significant evidence of proteoglycan loss in articular cartilage of Acan1$^{CreER}$, IKK2ca$^{ki/ki}$ (IKK2ca$^{acan}$) mice compared to tamoxifen-treated control mice (Fig. 3h; arrows). Gene expression in articular cartilage of IKK2ca$^{acan}$ mice knee joints displayed increased expression of catabolic genes such as IL-6, MMP13, ADAMTS4 and MCP-1 compared with wild-type (WT) mice (Fig. 3i–m). Thus, constitutive activation of IKK2/NF-κB in chondrocytes is sufficient to cause OA-like joint damage and resembles OA changes induced by MLI. This does not suggest that chondrocytes are the only source of inflammatory stimuli generated in the joint, as other cells such as synovial cells and macrophages are also contributors, but the inflammatory response of chondrocytes to such stimuli is critical for cartilage degradation.

**LDHA inhibition reduces catabolic activity by IκB-ζ protein.** NF-κB has many important physiological functions, making chronic or systemic NF-κB inhibition detrimental due to wide-spread side effects. We instead targeted the metabolic shift using the LDHA inhibitor FX11[44] to inhibit the final step of the aerobic glycolytic pathway and observed significant inhibition of inflammatory response genes such as IL-6 and MMP13 (Fig. 4a, b). Inhibition of LDHA was not toxic to cells and did not affect cell viability (Supplementary Fig. 3A, B). Supporting these findings, LDHA inhibition by FX11 did not significantly decrease cellular ATP levels as expected or change ECAR (Supplementary Fig. 3C, D). It is likely that there is partial inhibition of LDHA,

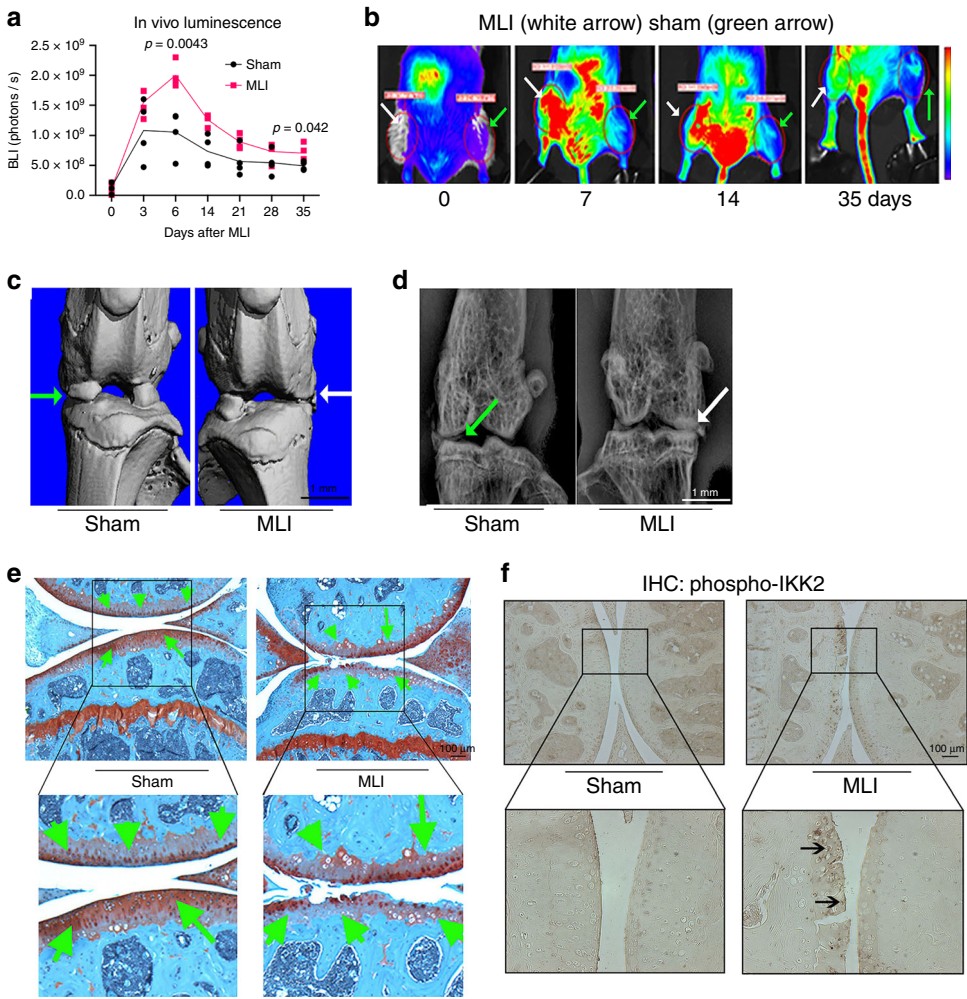

**Fig. 2 NF-κB is activated in chondrocytes of PTOA knee joints.** MLI surgery was performed in 12-week-old NF-κB-luciferase reporter mice. Right knee joints were operated for MLI surgery and left knee joints were used for sham surgeries ($n = 4$ each group). **a** Bioluminescence imaging after MLI and sham surgery showed increased NF-κB activation in the MLI surgery joint (right leg) compared to sham-operated knee joint in the same animals over 5 weeks. Graphs display mean ± S.D for $n = 4$ mice per side. Two-tailed student's $t$-tests were performed at each timepoint comparing the two groups with $p$-values indicated in figure. **b** Representative images of bioluminescence imaging from the same mouse at various times after surgery. **c** Representative μCT image and **d** X-ray image from one mouse showed joint damage, loss of synovial space and osteophyte formation in MLI surgery operated knee joint (white arrow). **e** Representative image of safranin-O staining of knee joints from same mouse shows loss of articular cartilage and proteoglycans after 8 weeks of MLI surgery compared to sham surgery. **f** Representative image from one mouse of IHC staining for p-IKK2 (arrows) in knee joints upon MLI surgery compared to sham surgery. **e–f** Scale bar = 100 μm.

and compensation by LDHB that allows for continued cell survival, as has been shown in other studies[45]. We then tested if LDHA inhibitor directly decreases NF-κB transcriptional activation by utilizing NF-κB-luciferase reporter chondrocytes and observed that surprisingly, LDHA inhibitor (24 h) did not alter NF-κB activation by IL-1β (Supplementary Fig. 3E).

In light of these observations, we hypothesized that NF-κB-induced LDHA activity reciprocally regulates expression of inflammatory response genes via targeting downstream products of NF-κB. A prime candidate was *Nfkbiz*, one of the most significantly upregulated genes in our RNA-seq dataset upon IL-1β stimulation (Fig. 1a), which encodes IκB-ζ, a pro-inflammatory protein that has been shown to be important in macrophages for production of a subset of inflammatory response genes[13,46,47]. We identified that gene expression of *Nfkbiz* was significantly elevated in joint articular cartilage at 4 weeks post MLI (Fig. 4c). Confirming that this finding is inflammation-mediated, mRNA expression of *Nfkbiz* upon treatment of chondrocytes with IL-1β showed a significant and rapid increase,

which was NF-κB dependent since IKK2 inhibitor treatment was able to significantly attenuate it (Fig. 4d). Protein levels of IκB-ζ, which is not expressed in untreated cells, were significantly increased upon treatment with IL-1β, while co-treatment with IKK2 inhibitor greatly reduced it, corroborating gene expression findings (Fig. 4e). Genetic deletion of *Nfkbiz* demonstrated that IκB-ζ is essential for the expression of a subset of inflammatory response genes such as IL-6 and MMP13 in chondrocytes, even when NF-κB signaling is intact (Fig. 4f–g). This highlights IκB-ζ as a critical player in the inflammatory response and OA cartilage degradation.

Based on the findings that FX11 reduced gene expression of catabolic genes independent of NF-κB activity, we proposed that inhibition of LDHA may decrease IκB-ζ expression. FX11 did not decrease gene expression of *Nfkbiz* induced by IL-1β (Fig. 4h) but significantly decreased IκB-ζ protein level (Fig. 4e). These observations together with our findings that chondrocytes under basal conditions express the *Nfkbiz* gene but not IκB-ζ protein, and that IL-1β is essential for IκB-ζ protein expression, suggested

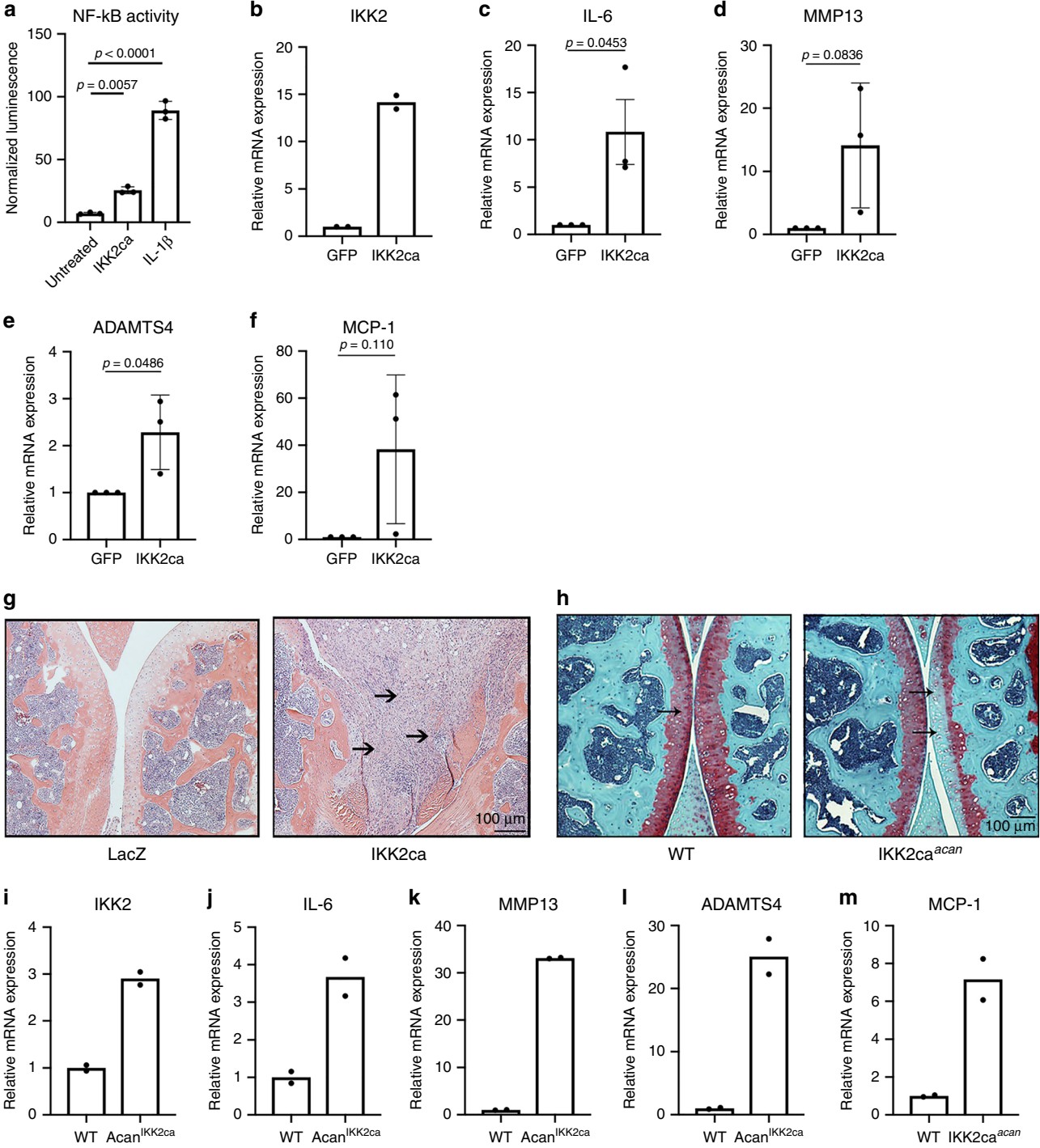

**Fig. 3 NF-κB activation in chondrocytes promotes catabolic changes. a** Primary chondrocytes isolated from RelA-Luc reporter mice were either treated with IL-1β (10 ng/mL) for 24 h or transduced with pMX-IKK2ca followed by luciferase assay, displaying NF-κB activation, normalized to protein levels. Bar graphs display mean ± S.D. of $n = 3$ wells. Data is representative of one of three independent experiments. **b–f** Primary chondrocyte isolated from wild-type mice were transduced with pMX-GFP or pMX-IKK2ca and cultured for 24 h. qPCR was performed to measure gene expression. Bars represent mean ± S.E.M. for $n = 3$ independent experiments. Two-tailed Student's $t$-test was utilized for statistical analysis, with $p$-values displayed in figure. **g** Six-week-old C57BL/6 mice were injected with Adeno-LacZ (left knee joint) and adeno-IKK2ca (right knee joint) ($n = 5$ males). Animals were sacrificed after 10 days of injection and knee joints were isolated for analysis. Representative image from one mouse of H&E staining of Ad-LacZ and Ad-IKK2ca injected joints reflects changes seen in all mice. **h** 12-week-old Agn1CreERT2, IKK2caki/ki (IKK2ca*acan*) and littermate controls animals ($n = 6$) mice were fed with tamoxifen diet (0.4 g/kg diet) for 2 months. At the end of the experiment, animals were sacrificed and knee joint tissue were harvested for further analysis. Representative image of Safranin-O staining from one mouse shows loss of articular cartilage and proteoglycans in IKK2ca*acan* mice compared to littermate controls. **i–m** Gene expression measurement from mRNA isolated from pooled articular cartilage of IKK2ca*acan* mice ($n = 3$) compared to control mice ($n = 3$), due to small size of tissue sample. Representative data from one experiment out of two, with bars representing mean of technical duplicates.

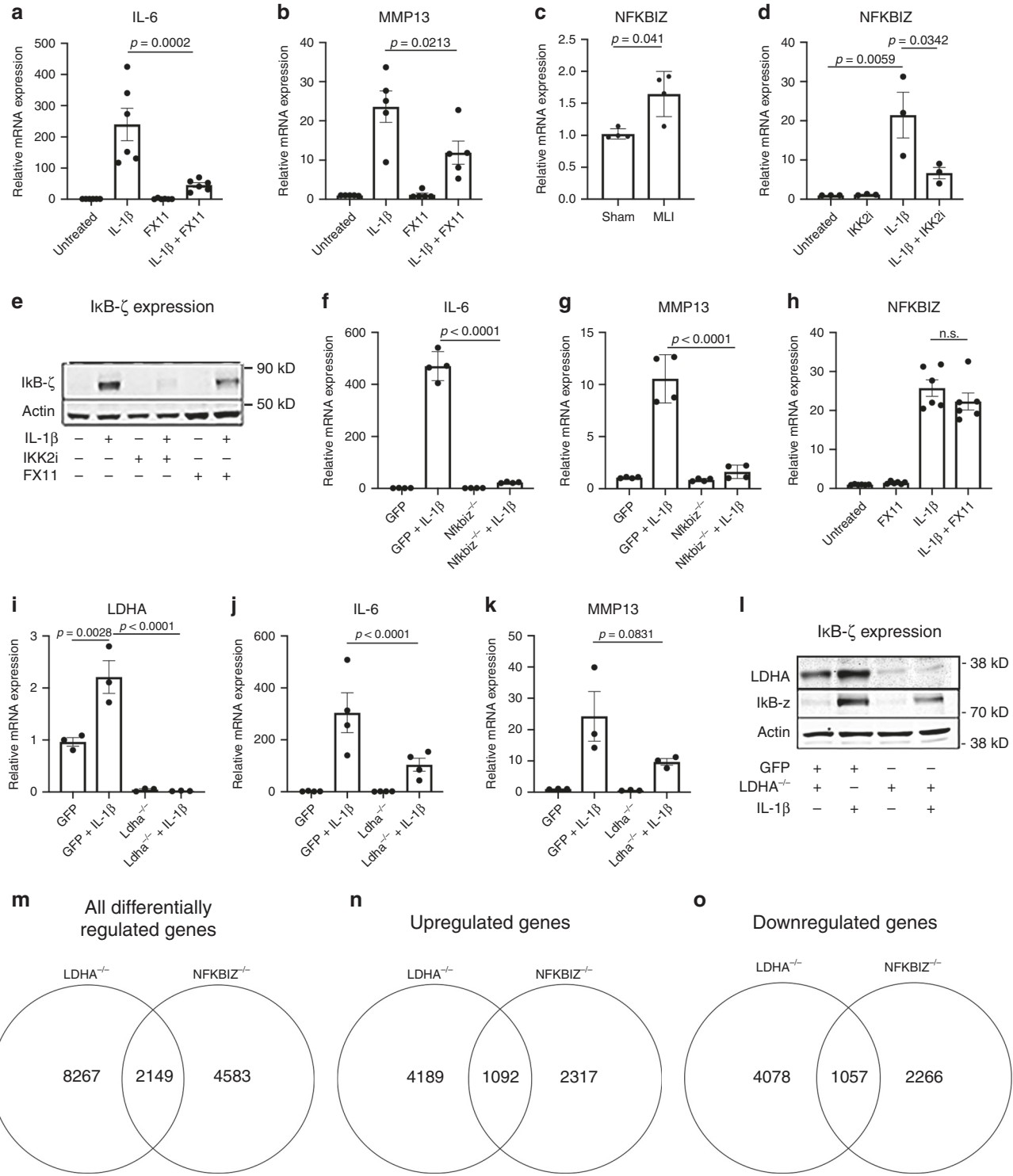

that IκB-ζ is regulated in a bi-modal manner transcriptionally and post-transcriptionally by IKK2 and LDHA, respectively. Proposing that IκB-ζ is likely regulated post-translationally, we displayed that IκB-ζ is targeted to the proteasome for degradation since use of proteasome inhibitor lead to accumulation of IκB-ζ even in the absence of inflammatory stimuli (Supplementary Fig. 3F; lanes 5, 7).

To confirm that inhibition of LDHA was responsible for the decreased expression of catabolic genes, and not due to an off target effect of FX11, LDHA was deleted from primary chondrocytes using $LDHA^{f/f}$ cells. LDHA deficient cells

($LDHA^{-/-}$) displayed a significant decrease in expression of IL-6 and MMP13, confirming that LDHA supports catabolic gene expression (Fig. 4i–k). In addition, $LDHA$ deletion lead to significant inhibition of IκB-ζ protein expression induced by IL-1β treatment (Fig. 4l). We then performed RNA sequencing to compare gene expression in LDHA and $Nfkbiz$ deficient to wild-type chondrocytes treated with IL-1β. We identified many overlapping genes in $LDHA$ deficient and $Nfkbiz$ deficient chondrocytes, further suggesting that much of LDHA's anti-inflammatory properties are IκB-ζ dependent (Fig. 4m–o). Of note, deletion of LDHA or use of FX11 did not significantly

**Fig. 4 LDHA regulates catabolic gene expression and IκB-ζ protein levels. a, b** Primary chondrocytes were treated with IL-1β (10 ng/mL) and/or FX11 (40 μM) for 24 h. Bar graphs represent qPCR data for IL-6 and MMP13 expression with error bars representing mean ± S.E.M. of $n = 5$ experiments. **c** Articular cartilage was isolated from knee joints of mice 4 weeks after undergoing sham or MLI surgery. Gene expression of *Nfkbiz* was measured by qPCR, with bars representing mean ± S.D. from $n = 4$ mice. Two-tailed student's *t*-test was performed. **d** Primary chondrocytes were treated with IL-1β and/or IKK2i (10 μM) for 24 h. Gene expression of Nfkbiz was measured by qPCR. Bars represent mean ± S.E.M. from $n = 3$ independent experiments. **e** Representative immunoblot for IκB-ζ from chondrocytes treated with IL-1β and/or IKK2i (10 μM) or FX11 (40 μM) for 24 h. **f–g** Primary *Nfkbiz*[f/f] chondrocytes were infected with adeno-GFP or adeno-cre (labeled Nfkbiz[−/−]), then treated with IL-1β (10 ng/mL) for 24 h. Gene expression of IL-6 and MMP13 was measured, with bars representing mean ± S.D. for $n = 4$ replicates per group, representative of one of three independent experiments. **h** Primary chondrocytes were treated with IL-1β and/or FX11 (40 μM) for 24 h. Gene expression of *Nfkbiz* was measured. Bars represent mean ± S.E.M. for $n = 6$ independent experiments. **i–k** Primary LDHA[f/f] chondrocytes were infected with adeno-GFP or adeno-cre (labeled LDHA[−/−]), then treated with IL-1β (10 ng/mL) for 24 h. Gene expression of LDHA, IL-6, and MMP13 was measured. Bars represent mean ± S.E.M. from $n = 3$ independent experiments for LDHA and MMP13, $n = 4$ for IL-6. **l** Representative image of immunoblotting performed for LDHA and IκB-ζ under the same conditions. **m–o** WT, LDHA[−/−], and NFKBIZ[−/−] chondrocytes were cultured with IL-1β treatment for 24 h and RNA sequencing was performed to compare gene expression to control chondrocytes. **m** The number of common, significantly regulated genes in both sets was counted. **n** The common genes significantly upregulated in both sets was counted. **o** The common genes significantly downregulated in both sets were counted. **a–b, d, f–k** One-way ANOVA followed by Tukey's multiple comparison's test was performed. *p*-values displayed in figure.

reduce lactate secretion (Supplementary Fig. 3G) or viability of chondrocytes, likely due to LDHB compensation. This confirms a previous report that deletion of both LDHA and LDHB is required in some cell types to eliminate lactate production[45,48]. We confirmed this proposition by using the LDH inhibitors GSK2837808A and sodium oxamate, which can act on both LDHA and LDHB. Indeed, both inhibitors were able to effectively reduce lactate production (Supplementary Fig. 3H). Similarly, deletion of LDHA alone was unable to significantly reduce lactic acid secretion (Supplementary Fig. 3I). We also observed that inflammatory stimulation had opposite effects on LDHB gene expression levels, with IL-1β treatment reducing LDHB gene expression (Supplementary Fig. 3J), yet residual activity of this enzyme supported lactate production. This is also supported by the fact that at the protein level, 24 h of IL-1β treatment did not decrease LDHB levels (Supplementary Fig. 3K). Various LDH inhibitors also did not affect protein levels, displaying the availability of LDHB to compensate.

**LDHA-NADH interaction propagates ROS induced by inflammation.** Next, to decipher the mechanism by which LDHA modulates the expression of IκB-ζ and the inflammatory response, we first determined that LDHA does not regulate mRNA stability of *Nfkbiz*, suggesting that IκB-ζ is likely regulated at or beyond the stage of translation (Supplementary Fig. S4A). This is in agreement with previous publications that suggest that IκB-ζ is not regulated at the level of mRNA stability[49]. Since LDHA's primary function is to generate lactate, we examined if lactate levels can affect IκB-ζ stability and/or the inflammatory response. Treatment of chondrocytes with lactate did not affect IL-6 and slightly increased MMP13 gene expression, but had no effect on IκB-ζ protein expression (Supplementary Fig. 4B–D). We also observed that treatment of cells with super-physiological amounts of pyruvate marginally decreased the inflammatory response but not IκB-ζ protein levels (Supplementary Fig. 4E–G). These results rule out lactate and pyruvate as major contributors to IκB-ζ.

Next, since LDHA inhibitor FX11 is a partial NADH analog[44,50], we surmised that the mechanism of action is dependent upon LDHA binding to NADH. Prior publications in cell-free systems showed that LDHA bound to NADH can promote electron donor activity of NADH to generate ROS species, thus creating a ROS chain reaction[51–53]. Using the DCFDA fluorescent probe, we show that IL-1β significantly increased intracellular ROS production in an NF-κB-dependent manner (Fig. 5a). More importantly, FX11 treatment abolished ROS production (Fig. 5a).

There are several sources of ROS in the cell, though the most likely sources upon inflammatory stimulation are either the activity of NADPH oxidase (NOX) or mitochondrial activity. NOX activity is dependent upon NADPH, generated via pentose phosphate activity, which we have shown to be increased in chondrocytes treated with IL-1β (Supplementary Fig. 1E). We show that NOX2 and NOX4 are expressed in chondrocytes, with NOX2 expression being upregulated following IL-1β treatment (Fig. 5b, c). Interestingly, DPI, a NOX inhibitor, or 6-ANA, a PPP inhibitor, greatly decreased the IL-1β-induced increase of ROS, indicating that NOX activity is a critical component of IL-1β-mediated ROS production (Fig. 5d). These results were further confirmed by the use of Vas2870, a highly specific NOX inhibitor (Supplementary Fig. 4H, I).

NADPH can also be consumed by ROS-reducing GSH regenerating enzymes. The results of GSH:GSSG measurements suggest PPP activity is likely increased to maintain GSH levels and attempts to balance out the pro-ROS effect of IL-1β (Supplementary Fig. 4J). FX11 causes a slight increase in GSH, suggest a lower state of oxidative stress, likely due to overall lower ROS in the cell.

We also measured levels of ROS generated in mitochondria and observed increased mtROS upon IL-1β treatment (Supplementary Fig. 4K), due to mitochondrial dysfunction displayed by the decrease in OxPhos observed in Fig. 1. This dysfunction likely contributes to increased NADH availability, as was previously published[54,55], further contributing to the propagation of ROS by LDHA. However, use of FX11 did not decrease mtROS production.

To confirm that the ROS-propagating function of LDHA is not mediated by its enzymatic activity, but rather through interaction with NADH, we utilized sodium oxamate, a pyruvate-analog LDHA inhibitor, which should not interfere with LDHA-NADH interaction. We show that this inhibitor had no effect on ROS (Supplementary Fig. 4L), though at these concentrations it significantly decreased lactate production (Supplementary Fig. 3H) confirming inhibition of LDHA enzymatic activity. Furthermore, we display that FX11 increased levels of NADH relative to NAD in the cell, supporting our hypothesis that FX11 displaces NADH from LDHA and stops the donation of electrons from NADH to free radicals (Fig. 5E). LDHA deletion also reduced the amount of ROS generated in the cell upon IL-1β stimulation (Supplementary Fig. 4M). Thus, we confirm that blocking the NADH binding site on LDHA is critical for ROS generation compared to the substrate binding site.

Finally, to rule out the potential role of the higher atmospheric oxygen in modulating metabolism and the inflammatory response,

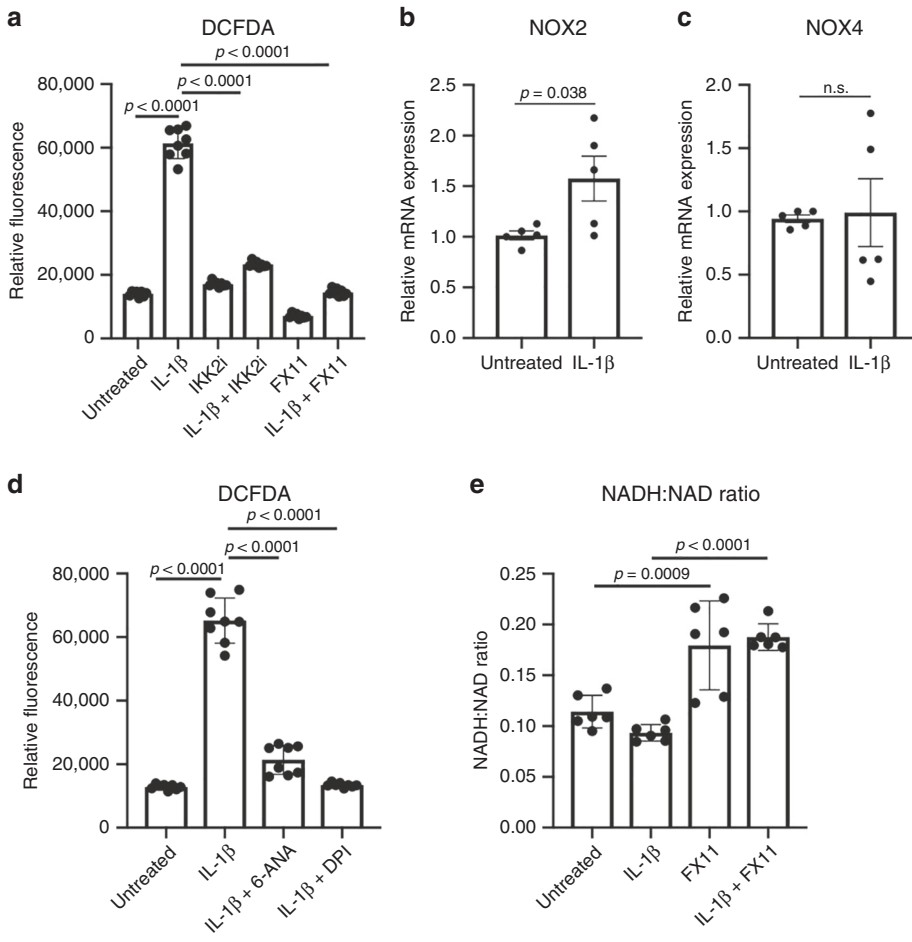

**Fig. 5 LDHA interaction with NADH propagates ROS. a** Primary chondrocytes were treated with IL-1β (10 ng/mL) in the presence or absence of IKK2i (10 μM) or FX11 (40 μM) for 24 h. DCFDA assay was performed and fluorescence was measured by microplate reader to determine intracellular ROS levels, displaying reduction in ROS levels. Results are representative of one experiment out of three, with bars representing mean ± S.D. for $n = 8$ wells. **b, c** Primary chondrocytes were treated with IL-1β (10 ng/mL) for 24 h. qPCR was performed to measure gene expression of NOX2 and NOX4. Bars represent mean ± S.E.M. for $n = 5$ independent experiments. **d** Primary chondrocytes were treated with IL-1β (10 ng/mL) in the presence or absence of 6-ANA (100 μM) or DPI (1 μM) for 24 h. DCFDA assay was performed and fluorescence was measured by microplate reader to determine intracellular ROS levels, displaying reduction in ROS levels. Results are representative of one of three independent experiments, with bars representing mean ± S.D. for $n = 8$ wells. **e** Chondrocytes were treated with IL-1β (10 ng/mL) for 36 h in the presence of absence of FX11 (40 μM). NAD$^+$ and NADH levels were measured in chondrocytes to calculate the NADH:NAD$^+$ ratio in biological replicates. Data are pooled from three experiments for $n = 6$ biological replicates, with error bars representing mean ± S.D. **a, d, e** One-way ANOVA was performed followed by Tukey's multiple comparison's test. **b, c** Two-tailed student's $t$-test was performed. $p$-values displayed in figure.

we wanted to determine if the same effects of inflammation and LDHA inhibition on ROS occur in hypoxic conditions. We observed that chondrocytes cultured and treated in hypoxia exhibited similar inflammatory responses to IL-1β (Supplementary Fig. 5A–C). However, there was increased expression of LDHA and production of lactate, suggesting a greater reliance upon anaerobic glycolysis as expected under hypoxia. In addition, ROS levels also changed with similar patterns as that observed in normoxic culture conditions, although there was slightly more ROS with IL-1β stimulation, likely due to increased LDHA activity with hypoxia (Supplementary Fig. 5E). FX11 was still highly effective however, highlighting that the effects of inflammation on metabolism and ROS seen are independent of environmental oxygen and originate from a cell intrinsic source. Finally, IκB-ζ expression in response to IL-1β also resembled that seen during normoxia (Supplementary Fig. 5F).

**ROS species stabilize IκB-ζ to promote inflammatory response.** We then hypothesized that inflammation-induced LDHA activity

propagates ROS production to cause oxidative stress, leading to stabilization of IκB-ζ, and that the protective effect of LDHA inhibitor (FX11) may be occurring via restraining damaging ROS species. Indeed, treatment of chondrocytes with ROS scavenging molecule N-acetyl-cysteine (NAC) was able to decrease protein levels of IκB-ζ induced by IL-1β and also decreased expression of IL-6 and MMP13 (Fig. 6a–c). We also confirmed this effect using DMSO, which has previously been shown to be a ROS scavenger and anti-inflammatory compound[56]. Increasing concentrations of DMSO reduced cellular ROS levels, IκB-ζ protein expression, and gene expression of IL-6 and MMP13 (Fig. S6A–D). Furthermore, all the treatments that were shown to decrease ROS such as Vas2870 and 6-ANA also potently decreased expression of IL-6 and MMP13 (Fig. S6E–H).

To further confirm the role of ROS in promoting IκB-ζ stability, chondrocytes co-treated with IL-1β and hydrogen peroxide showed increased IκB-ζ protein compared to IL-1β alone (Fig. 6d). In addition, hydrogen peroxide co-treatment with IL-1β increased expression of IL-6 and MMP13, both IκB-ζ

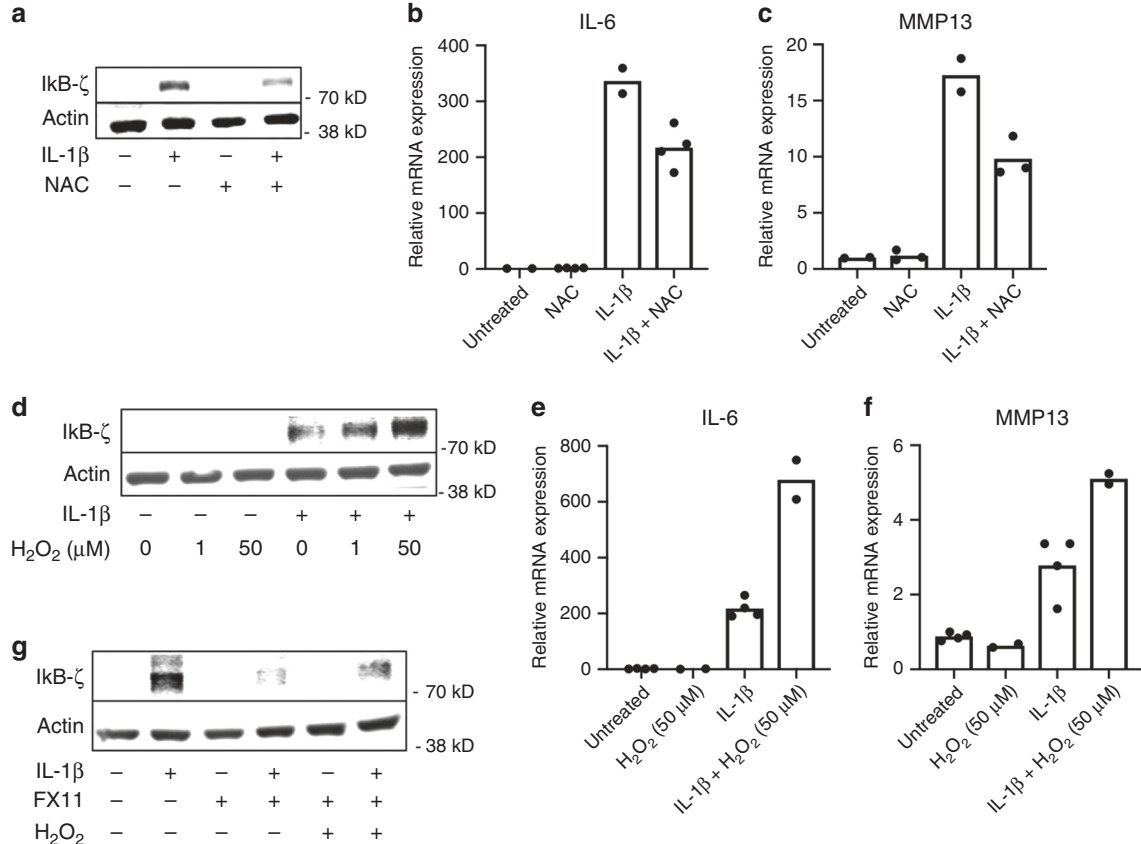

**Fig. 6 ROS regulates stability of IκB-ζ protein. a** Chondrocytes were co-treated with IL-1β (10 ng/mL) and N-acetylcysteine (3 mM) for 24 h. Western blotting was performed for IκB-ζ. **b, c** Gene expression analysis for IL-6 and MMP13 was performed by qPCR. Results are representative of two out of three independent experiments, due to large biological variation in third experiment with similar trends. Bars represent mean of $n = 2$ independent experiments. **d** Chondrocytes were treated with increasing concentrations of Hydrogen peroxide in the presence or absence of IL-1β for 24 h. Western blotting was performed for IκB-ζ. Image representative of one of three experiments. **e, f** Gene expression of IL-6 and MMP13 was measured by qPCR. Results are representative of one experiment out of three displaying similar trends but with large biological variations in gene expression. Bars represent mean of technical replicates from one experiment. **g** Chondrocytes were then treated with FX11 (40 μM) in the presence or absence of H₂O₂ (50 μM) and IL-1β (10 ng/mL) for 24 h. Western blotting was performed for IκB-ζ. Image representative of one of two experiments.

responsive genes (Fig. 6e, f). However, hydrogen peroxide alone had no effect on IκB-ζ protein, indicating that NF-κB activation, metabolic shift and ROS generation are critical parallel processes for IκB-ζ expression and stabilization. Furthermore, hydrogen peroxide co-treatment partially reversed the inhibitory effect of FX11 on IκB-ζ in chondrocytes by increasing protein levels relative to FX11 alone (Fig. 6g). Addition of hydrogen peroxide was also able to reverse the inhibitory effect of LDHA deletion or FX11 on IL-6 gene expression (Supplementary Fig. 6I, J), supporting that the effect of LDHA inhibition was ROS-mediated. Furthermore, sodium oxamate, which does not interfere with NADH binding, had little effect on ROS or IL-6 gene expression induced by IL-1β, or IκB-ζ protein levels (Supplementary Fig. 6K–M). However, GSK2837808A was able to markedly reduce the expression of IL-6, possibly because it is also a partial NADH analog (Supplementary Fig. 6N).

**Deletion of LDHA in chondrocytes is protective against OA.** To validate the functional significance of our biochemical findings, we generated a mouse model for chondrocyte-specific deletion of *Ldha* (*Acan*$^{\Delta Ldha}$), which are predicted to decrease ROS generation in response to inflammatory stimulation as well as decrease IκB-ζ protein expression, thus protecting mice from OA. Gene deletion was induced by feeding tamoxifen prior to performing MLI surgery. Safranin-O staining of the knee joints

displayed improved cartilage integrity in the MLI joints of *Acan*$^{\Delta Ldha}$ (Fig. 7a, lower panels; black arrows) animals compared with control MLI mice, indicating protection, while sham joints did not show significant differences between the groups (Fig. 7a; upper panels). Blinded OARSI scoring was performed to measure the severity of OA in the safranin-O stained sections, displaying significantly decreased OA severity with LDHA deletion (Fig. 7b). Furthermore, IHC staining for MMP13 in these joints displayed reduced MMP13 levels in the MLI joints of *Acan*$^{\Delta Ldha}$ animals compared to MLI joints of control animals (Fig. 7c), and the number of hypertrophic chondrocytes, which are indicative of damaged cartilage, appeared visually less in MLI sections of *Acan*$^{\Delta Ldha}$ animals.

**LDHA inhibition reduces catabolic response of HAC.** We then sought to determine the translational significance of our findings. We compared the gene expression profiles between well-preserved lateral and highly damaged medial human knee articular cartilage, confirmed by radiographic scoring and visual observation during surgery, from patients who underwent TKA. We observed a significant increase in LDHA (Fig. 8a) and G6PD2 (Fig. 8b) gene expression in damaged cartilage, indicating that human articular chondrocytes (HAC) in OA likely also undergo a metabolic reprogramming. In addition, damaged OA cartilage samples have elevated MMP13 (Fig. 8c) and NFKBIZ (Fig. 8d)

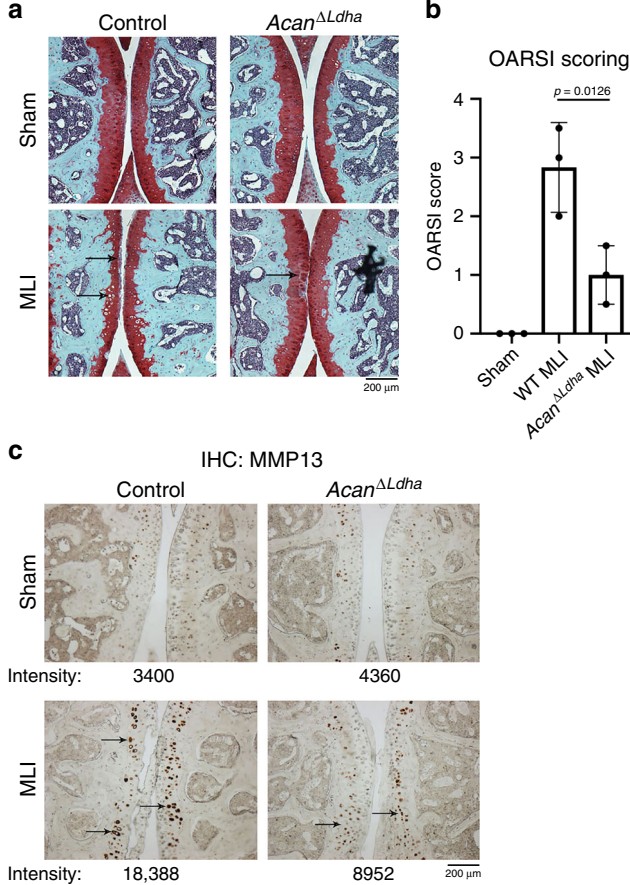

**a**

Control *Acan^{ΔLdha}*

Sham

MLI

200 μm

**b**

OARSI scoring

*p* = 0.0126

Sham WT MLI *Acan^{ΔLdha}* MLI

**c**

IHC: MMP13

Control *Acan^{ΔLdha}*

Sham

Intensity: 3400 4360

MLI

Intensity: 18,388 8952 200 μm

**Fig. 7 Deletion of LDHA in chondrocytes in vivo is protective against OA.**
**a** MLI surgery was performed on 10-week-old *Acan^{Δ/Ldha}* male mice placed on tamoxifen diet to induce recombination. Sham surgery was performed on contralateral leg and used as control. 10 weeks post surgery mice were sacrificed and joints were collected for histology. Safranin-O staining was performed of the MLI and sham joints. **b** OARSI scoring of two safranin-O stained sections per mouse were averaged ($n = 3$ mice for control, $n = 3$ mice for *Acan^{Δ/Ldha}*). Bars represents mean ± S.D. from one experiment out of two comparing littermates. One-way ANOVA followed by Tukey's test multiple comparisons test was utilized for statistical analysis. **c** IHC was performed for MMP13, with arrows pointing to examples of stained chondrocytes. Representative images are displayed with pixel intensity of stain quantified under each image. **a, c** Scale bar = 200 μm.

gene expression. To further validate our findings, we observed that primary HAC isolated from knee joints underwent similar metabolic changes when treated with IL-1β, with increased glycolytic activity and decreased peak oxygen consumption rate (Fig. 8e, f). We then performed IHC in human OA cartilage compared to healthy donors and observed increased staining for IκB-ζ protein in the OA cartilage (Fig. 8g, arrows).

Next, we validated our results utilizing HAC, treated with IKK2i or FX11 in the presence or absence of IL-1β. Treatment with IL-1β increased gene expression of *NFKBIZ*, which was significantly reduced with IKK2 inhibitor, but not with FX11 (Fig. 8h), replicating murine data (Fig. 4h). In addition, treatment with IKK2 inhibitor was highly effective at reducing catabolic enzyme and inflammatory cytokine production by inhibiting the NF-κB pathway (Fig. 8i, j). FX11 had a similar inhibitory effect on catabolic gene expression (Fig. 8i, j) as it did in murine chondrocytes by reducing IL-6 and MMP13 expression (Fig. 4a, b), identifying LDHA as a potential therapeutic target in human OA patients as well.

## Discussion

This work highlights several findings that have important implications for understanding the intricate relationship between the inflammatory and metabolic responses of OA chondrocytes. We conclude that, (a) Inflammation shifts cell metabolism towards aerobic glycolysis in an NF-κB-mediated manner, (b) LDHA has a pro-ROS forming function in chondrocytes during inflammatory states, (c) Inhibition of LDHA activity has potent anti-inflammatory and anti-catabolic properties via increasing degradation of IκB-ζ by the proteasome.

Unlike other joint diseases, such as RA, that have a significant pathogenic contribution from immune and other joint synovial cells, OA is primarily driven by the generation of catabolic products from articular chondrocytes in response to biomechanical and inflammatory stimuli. Here, we present definitive data that chronic NF-κB activation is one of the major drivers of pathological changes in OA. NF-κB can be activated in chondrocytes due to insults throughout life such as mechanical stress, injury, and the presence of inflammatory cytokines emanating from aging, metabolic disease and other causes to promote catabolic changes. Cartilage degradation further increases mechanical stress and injury to the joint, exacerbating inflammatory stimuli in the synovial space to perpetuate NF-κB activity and generate a vicious positive feedback cycle.

However, chronic systemic inhibition of NF-κB is challenging in humans due to undesired side effects. Similarly, inhibition of individual cytokines, such as IL-1β, using biologics has failed in the past for treating OA since it does not block the myriad of other potential inflammatory stimuli[57]. We instead explored a paradigm by which inflammation and metabolism are reciprocally regulatory. IL-1β treatment caused metabolic reprogramming, mimicking the Warburg effect and expanding upon some previous reports studying chondrocyte metabolism[58–60]. Other gene expression sets from articular chondrocytes treated with IL-1β or OA chondrocytes also display similar alterations in metabolic enzyme expression[61,62], confirming metabolic changes as a conserved aspect of OA. We highlight the metabolic shift is a potential target for treating OA and identify a non-metabolic function of LDHA that is critical for modulating the inflammatory response, independent of its canonical lactate-producing role. Chan et al. showed elegantly that LDHA binding to NADH in a cell-free system can greatly increase the rate of oxidation and free radical generation, causing a chain reaction that was dependent upon superoxide as the initiating factor[52,63,64]. It is likely that the interaction of NADH with the Rossman fold domain of LDHA[65,66] allows for faster electron donation from NADH to oxygen-containing compounds through its catalytic activity, as well as by thermodynamic stabilization of free radical intermediates. A similar finding was reported by another group studying the role of LDHA[67], validating our own results. However, the impact of this finding in biological systems and its implications for disease has not been well studied. While this work suggests that NADH is a pathological factor, it is likely only so during inflammatory states. During inflammation, elevated glycolysis leads to increased NADH levels, which is not consumed due to the decrease in ETC activity. Overall, this high-energy NADH excess can donate electrons to oxygen-containing compounds in addition to pyruvate. We propose however that this mechanism is only partially responsible for the effect of FX11 on inflammation since prior studies have shown that FX11 can also have some effects on mitochondria[44], which may explain why deletion of LDHA did not have as dramatic of an effect on ROS as FX11 treatment. More surprisingly, LDHA deletion or inhibition had little effect on chondrocyte viability or lactate levels, demanding further research into the role of other LDH isoforms during inflammatory states.

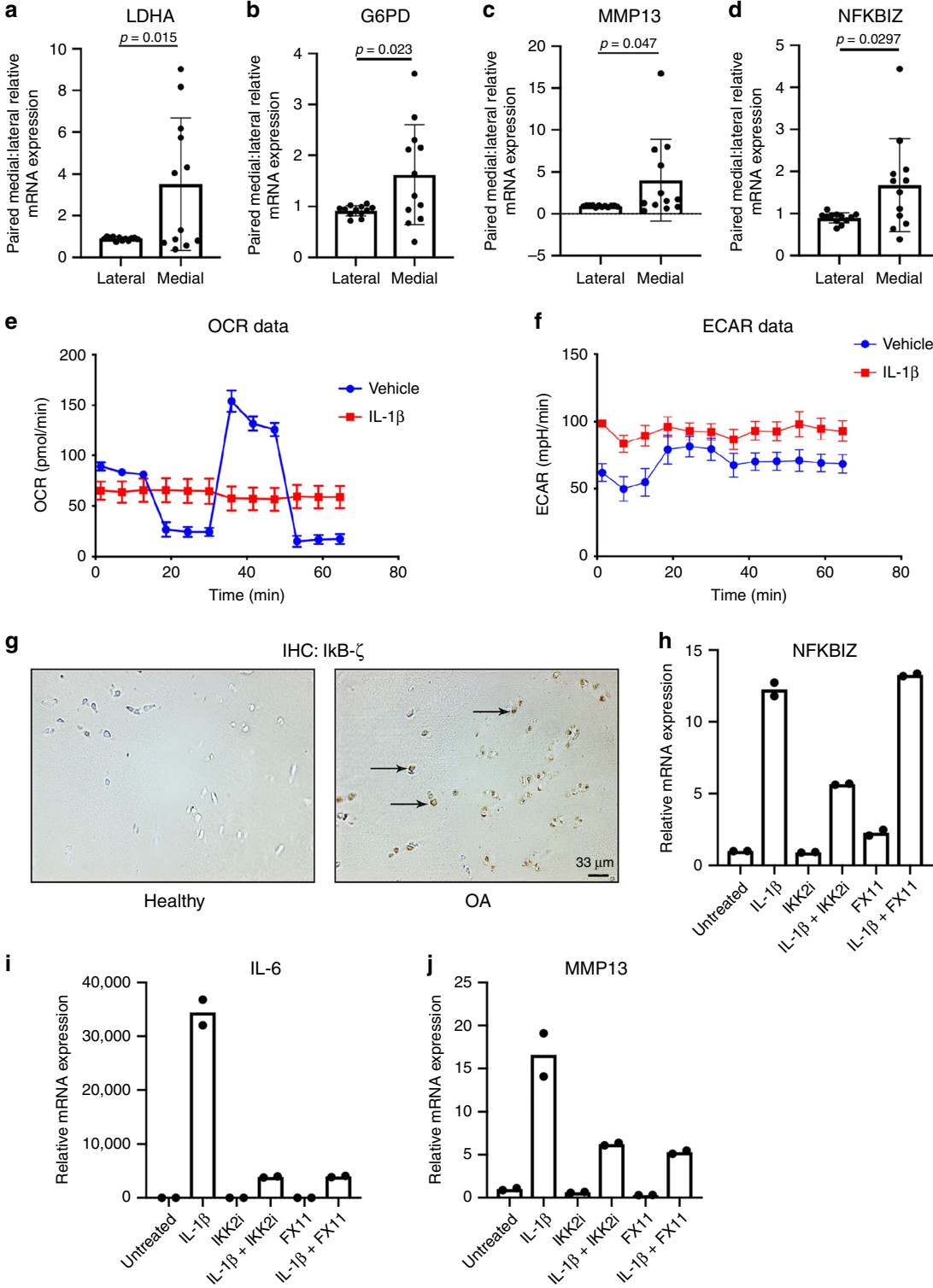

Mechanistically, we provide evidence that IκB-ζ plays an important part in the metabolic-inflammatory axis and is, at least partially, responsible for the pro-inflammatory effects of LDHA-induced ROS. IκB-ζ is an inflammatory mediator that we display is bi-modally regulated by NF-κB and ROS, transcriptionally and post-transcriptionally, respectively. The process begins with transcriptional expression of *Nfkbiz* by IKK2/NF-κB followed by stabilization of its protein product, IκB-ζ, in ROS-dependent manner. We display that IκB-ζ is redox-sensitive protein essential for the expression of a subset of inflammatory response genes, such as IL-

1β, MMP13 and IL-6, which are well established culprits of OA cartilage loss, a finding supported by recent evidence[68] and warrants discovery of direct inhibitors of IκB-ζ activity. One of the limitations of this study was the use of fluorescent probes such as DCFDA to study ROS production. The use of redox sensors would offer much more fine-tuned analysis of ROS species. Further elucidating this redox mechanism through state-of-the-art genetic models and redox measurement techniques will be the focus of future work, especially given the extensive evidence that oxidative stress is a prevalent factor in OA pathogenesis[31,69].

**Fig. 8 LDHA inhibition is efficacious in human articular chondrocytes.** Medial and lateral cartilage samples were obtained from knee articular cartilage of patients undergoing TKA ($n = 12$). Medial regions of articular cartilage were more severely damaged upon radiographic and visual analysis by surgeon, while lateral regions were healthier regions of cartilage with little signs of OA. **a–d** LDHA, G6PD2, MMP13, and NFKBIZ gene expression was measured in OA cartilage by qPCR. Gene expression was normalized to actin. Gene expression is displayed as fold change in medial cartilage sample relative to paired lateral cartilage sample. Bars represent mean ± S.D. for $n = 12$ patient samples. **e, f** Primary human knee articular chondrocytes were treated with IL-1β (10 ng/mL) for 24 h. Seahorse was performed to measure ECAR and OCR as previously stated. Bars represent mean ± S.D. for five wells. Data is representative of one out of two experiments. **g** IHC was performed for IκB-ζ in healthy and OA human cartilage (arrows). Representative images of $n = 3$ samples are displayed. **h–j** Chondrocytes were isolated from human knee articular cartilage. Cells were treated with IL-1β (10 ng/mL) for 24 h in the presence or absence of IKK2i (10 μM) or FX11 (40 μM). Gene expression of NFKBIZ, IL-6, and MMP13 was measured by qPCR, with bars representing mean of technical duplicates. Results are representative of one out of two independent experiments with similar trends. **a–d** Two-tailed student's $t$-test was performed. $p$-values indicated in figure.

Studies analyzing synovial fluid displayed that OA patients have elevated levels of lactic acid in the absence of sepsis, further supporting the notion that LDHA is likely a pathogenic player in human OA[70], which we confirmed using human OA tissue samples. Our work has important translational implications given that targeting LDHA may be very safe in humans. Humans who have nonfunctional LDHA mutations are often asymptomatic or have symptoms upon severe exertion[71,72], suggesting that inhibition of LDHA may not pose a significant detrimental effect to healthy cells. It is possible that with specific targeting of FX11 or NF-κB inhibitors to the joint, the progression of OA can be halted by preventing chondrocyte-mediated inflammatory response. Overall, our results identify LDHA-mediated ROS production as a therapeutic target for OA.

## Methods

**Animal models**. IKK2ca (IKK2ca^f/f^), LDHA floxed (Ldha^f/f^) (gift from Dr. Fanxin Long) on a C57BL/6 background were crossed with Aggrecan-cre–ERT2 mice (Stock No. 09148, Jackson Labs) to generate Agn1^CreERT2^,IKK2ca^ki/ki^ (IKK2ca^acan^) mice, Acan^ΔLdha^, on a C57BL/6 background, respectively. Only male mice were utilized for these experiments in order to maintain comparability to the MLI model, which is generally performed on male mice. All mice were fed with tamoxifen chow (Teklad Custom Diet, Envigo) to induce genetic recombination in inducible Aggrecan-cre-ERT2 mice, with groups separated into cre-positive and cre-negative animals. NF-κB-GFP-Luciferase reporter mice were purchased from Jackson Labs. IKK2 floxed, LDHA floxed, and *Nfkbiz* floxed mice (^68^RBRC06410, Riken, Japan) were used for isolating primary chondrocytes from newborn pups. NF-κB-GFP-Luciferase reporter mice (Stock No. 027529, Jackson Labs) were used in vivo bioluminescence imaging with MLI and for isolating primary chondrocytes for measuring NF-κB activity. Wild type (WT) mice were generated by breeding C57BL/6 animals. All experiments were performed using littermate controls. Mice were housed at the Washington University School of Medicine barrier facility at five or less per cage at 24–26 °C with humidity ranging between 30–60% with 12 h light/dark cycles switching at 6 p.m. All experimental protocols were carried out in accordance with the ethical guidelines approved by the Washington University School of Medicine Institutional Animal Care and Use Committee.

**Cell culture**. For all chondrocyte experiments unless otherwise stated, primary chondrocytes were isolated from the sterna and ribs of P1-P3 mice without consideration for sex using sequential digestion with pronase (2 mg/mL, PRON-RO, Roche) at 37 degrees, followed by collagenase D (3 mg/mL, COLLD-RO, Roche) two times at 37 degrees, and cultured in Dulbecco's modified Eagle medium (DMEM; Life Technologies, USA) containing 10% fetal bovine seru (FBS) and 1% penicillin/streptomycin (15140122, ThermoFisher) and plated in tissue culture plates, based on previous protocol[73]. For experiments, cells are treated with recombinant mouse IL-1β (211-11B, Peprotech) at 10 ng/mL, LDHA inhibitor FX11 (427218, EMD Millipore), Sodium Oxamate (O2751, Sigma), GSK2837808A (MedChem Express), IKK2 Inhibitor SC-514 (SML0557, Sigma), Diphenyleneiodinium (D7008, Sigma), 6-aminonicotinamide (A68203, Sigma), Vas2870 (SML0273, Sigma) or hydrogen peroxide (323381, Sigma). IKK2ca expression experiments were performed under similar conditions without treatment of IL-1β following retroviral transduction. Human chondrocytes were isolated from post surgery discarded human articular cartilage samples isolated by enzymatic digestion. Cells were cultured in DMEM:F12 (Life Technologies, USA) containing 10% FBS, 1% penicillin/streptomycin (15140122, ThermoFisher) and Ascorbic acid. To generate retroviral particles, Plat-E cells were used. Plat-E cells were cultured in DMEM supplemented with 100 units/mL penicillin/streptomycin and 10% FBS (v/v).

**Meniscal-ligamentous injury**. Meniscal-ligamentous injury (MLI) surgery was utilized to induce OA in mice[74]. In this procedure, medial meniscus was surgically removed from the joint without disrupting patella or other ligaments. Sham surgery was performed on the contralateral joint in which a similar incision is made on the medial side without removal of the meniscus. After 10 weeks, mice are sacrificed and joints were collected for histology. MLI was performed in 10–12-week-old male NF-κB-GFP-Luciferase reporter mice to generate and analyze NF-κB activation in OA model. Acan^ΔLdha^ mice underwent MLI surgery to study the role of LDHA deletion on OA development. Mice were anesthetized with ketamine according to the approved animal protocol. To analyze the effect of IKK2 inhibitor VIII (HY-13060, Medchem Express, Monmouth Junction, NJ) on OA development, MLI surgeries were performed in 12-week-old C57BL/6 WT mice. The mice were injected with IKK2 inhibitor VIII (2 mg/kg, IP) every alternate day for 5 weeks. OARSI scoring was performed by blinded reviewer based on safranin-O stained images observing articular cartilage structure and integrity. For isolation of articular cartilage from these mice to study gene expression changes in the MLI compared to sham joints, mice were sacrificed 4 weeks post surgery and articular cartilage was scraped from the surface of femur and tibia of either sham or MLI joints under dissection microscope. Cartilage was then immediately placed in Trizol (15596026, ThermoFisher) and mRNA was isolated.

**Adenoviral injection in joints**. Adeno-LacZ and Adeno-IKK2ca viral particles were purchased from Viraquest, Inc., North Liberty, IA. $1 \times 10^9$ particles ($3–4 \times 10^7$ p.f.u.) Adeno-LacZ and Adeno-IKK2ca particles were injected in left and right knee, respectively, of 6-week-old C57BL/6 mice. The mice were sacrificed 10 days after injection. The joint tissues were collected and processed for histology and immunohistochemistry.

**Histology and immunochemistry**. At the end of experiments, mouse long bones were harvested keeping knee joints intact and fixing in 10% neutral buffered formalin for 24 h at room temperature followed by decalcification in Immunocal (StatLab, McKinney, TX) for 3 days with fresh Immunocal changed every 24 h. Tissues were processed, embedded into paraffin, and sectioned 5 μm thick then stained with Hematoxylin-Eosin or Safranin-O to visualize cartilage and bone. For immunohistochemistry, sections were deparaffinized and rehydrated using three changes of xylenes followed by ethanol gradient. Antigen retrieval in murine sections was performed by boiling samples in Citrate buffer (pH 6.0) at high pressure, followed by quenching of endogenous peroxidase activity by incubation in 3% $H_2O_2$ for 15 min at room temperature. Antigen retrieval in human cartilage sections was performed using proteinase K incubation for 15 min followed by peroxide quenching. Sections were then blocked using DAKO solution with background reducing components (S302281, Agilent, Santa Clara, CA) for 1 h at room temperature. Sections were incubated overnight at 4 degrees with anti-p-IKK2 (ab5915, Abcam, Cambridge, UK), anti-MMP13 (Ab39012, Abcam, Cambridge, UK), or anti-IκB-ζ (NBP1-89835, Novus, Colorado, US) antibody at a 1:200 dilution. Sections were rinsed in phosphate-buffered saline (PBS) several times followed by addition of 1:500 dilution of biotinylated secondary antibody (BP-1100, Vector Biolabs, Burlingham, CA) for 1 h. Post secondary antibody incubation, sections were washed with PBS-T several times followed by incubation with streptavidin-HRP (2 μg/mL) for 20 min. After extensive washing with PBS, sections were developed using DAB peroxidase kit (SK4100, Vector Biolabs, Burlingham, CA), with development on each slide standardized to the same amount of time. Images were quantified using ImageJ (RRID: SCR_003070) by converting images to greyscale, followed by setting a threshold value and measuring and summing total pixel intensity above the threshold.

**Bioluminescence imaging**. Bioluminescence imaging (BLI) was performed with the IVIS 50 imaging system (Xenogen/PerkinElmer) at Washington University Molecular Imaging Core. 5 min before anesthesia via isofluorane inhalation, mice were injected intraperitoneally with a luciferin solution (150 mg/kg). Luciferase activity was acquired by 10 s of photon imaging, two mice at a time. Mice were continuously imaged before and each week after surgery (for 5 weeks). All images

were analyzed by Living Image 3.0 software (RRID: SCR_014247, PerkinElmer) for luminescent intensity of the knee areas of the MLI side compared to the Sham side.

**Micro-CT and X-ray analysis**. Intact knee joint from different animals were isolated, fixed overnight in 10% neutral buffered formalin (HT501128, Sigma) followed by washing with phosphate buffer saline (PBS) three times and transfer to 70% ethanol (v/v). Bones were then scanned using Scanco Medical micro-CT systems (Scanco, Wayne, PA, USA) at the core facility at the Musculoskeletal Center at Washington University in St. Louis (St. Louis, MO). Briefly, images were scanned at a resolution of 20 μm, slice increment 20 μm, voltage 55 kV, current 145 μA and exposure time of 200 ms. After scanning the knee joint area, three-dimensional images were constructed. X-ray analysis of isolated knee joints were performed using Faxitron UltraFocus 100 on automatic settings and at x5 magnification.

**Transfection and viral infection**. GFP and IKK2ca were cloned in retroviral pMX constructs. For exogenous expression, relevant pMX-constructs were first trans-fected into Plat-E cells using Xtreme Gene 9 (XTG9-RO, Roche), followed by collection of virus-containing media for 2 days. Virus-containing media was later used to transduce primary chondrocytes by replacing culture media with virus-containing media in the presence of polybrene (5 ng/mL, TR-1003, Sigma) for 24 h. Media was then changed to DMEM with 10% FBS after 24 h.

For adenoviral infection, chondrocytes were treated with adenoviral-eGFP (VVC-U of Iowa 4) or adenoviral-Cre-eGFP (VVC-U of Iowa 1174) (University of Iowa Viral Vector Core) at an MOI of 10 in the presence of polybrene (5 ng/mL, Sigma) in media containing 5% FBS for 24 h. After 24 h, media was changed to DMEM containing 10% FBS. For DCFDA experiments using floxed cells, adenoviral-LacZ (VVC-U of Iowa 3554) and adenoviral-Cre (VVC-U of Iowa 5) (University of Iowa Viral Vector Core) were used to avoid fluorescent overlap of DCFDA and GFP.

**Western blot analysis**. Total cell lysates for protein analysis were prepared by scraping cells in 1x Cell Lysis Buffer (Cell Signaling Technology, Danvers, MA, USA) containing protease/phosphatase inhibitor (ThermoFisher Scientific, Halt Protease Phosphatase Inhibitor Cocktail). Blotting was performed using primary antibodies for LDHA (Cat# PA5-27406, Invitrogen, 1:1000 dilution), LDHB (14824-1-AP, Proteintech, 1:1000 dilution), IκB-ζ (Cat# 14-16801-82, Invitrogen, 1:1000 dilution), and Actin (Cat# A228, Sigma, 1:5000 dilution). Protein concentration was determined by BCA assay (Pierce) and equal amounts of protein were loaded onto sodium dodecyl sulfate–polyacrylamide gel electrophoresis gel. After wet transfer to nitrocellulose membrane using BioRad Transfer system using 1x transfer buffer, and blocking with 5% BSA in PBS-Tween (0.1% v/v) (PBST) for 1 h at room temperature, membranes were probed with specific primary antibody primary antibodies diluted in 5% BSA in PBST overnight at 4 °C. Membranes were washed three times with PBST and probed with secondary antibodies from LI-COR (Odyssey Imager; donkey anti-rabbit/IRDye, 800CW/anti-goat/IRDye 800CW, anti-mouse IRDye 680RD, Donkey anti-rat/IRDye 800CW, 1:10,000 dilution) for 1 h at room temperature. Membranes were then washed three times with PBST and scanned using LI-COR Odyssey Imager (LI-COR Biosciences, Lincoln, NE, USA) at low resolution using Licor software (RRID: SCR_014579).

**Extracellular lactic acid measurement**. Chondrocytes were cultured for 1 day with IL-1β treatment (10 ng/mL) with appropriate experimental conditions in 24-well plates containing 500 μL of DMEM containing 10% FBS. Supernatant media was collected and centrifuged to separate cell debris and floating cells. Supernatant was used immediately or stored at −80° C until lactic acid assay was performed to measure secreted lactate in the media using a 1:20 dilution (Cat# 1200011002, Eton Biosciences, L-Lactate Assay Kit I). Unconditioned DMEM with 10% FBS was used as a control for subtracting background.

**qPCR analysis**. Trizol (Sigma) was added to samples to isolate mRNA from cell culture samples and human articular cartilage samples. Chloroform was added at a ratio of 0.2:1 to Trizol to samples, followed by centrifugation at $12,000 \times g$ for 15 min. Aqueous layer was isolation and equal amount of 70% ethanol was added. RNA was then isolated from this fraction using PureLink RNA mini kit (Cat# 12183025, Ambion, Grand Island, NY, USA) and complementary DNA (cDNA) was prepared using High-Capacity cDNA Reverse Transcription kit (Cat# 4368814, Applied Biosystems). qPCR was carried out on BioRad CFX96 real time system using iTaq universal SYBR green super-mix (Cat#1725120, BioRad, Hercules, CA, USA). mRNA expression was normalized using actin as a housekeeping gene. Full list of primers is listed in Supplementary Table 1.

**Measurement of cellular metabolism by seahorse**. Primary chondrocytes were plated in Seahorse XF96 plates at 50,000 cells per well and treated with IL-1β (10 ng/mL) in the presence or absence of IKK2 inhibitor or FX11. After 72 h, Seahorse assay was performed. For glycolysis stress test, cells were serum starved for 1 h in glucose-free media containing treatments, and measurement of ECAR and OCR was performed prior to and after sequential addition of glucose,

oligomycin and 2-DG with measurements performed every 5 min. For MitoStress test, cells were incubated in glucose-containing media for 1 h containing treatments and measurements were performed every 5 min prior to and after sequential addition of oligomycin, FCCP and Rotenone/Antimycin A. Data was analyzed using Wave software.

**Measurement of intracellular ATP**. Primary chondrocytes were plated in 96-well plates at $5 \times 10^4$ and treated with IL-1β for 24 h. Lysates were collected and processed according to luminescence-based ATP assay kit (Cat#K255, Biovision, ADP/ATP ratio Assay kit). Assay was performed in 96-well plate. Luminescence was measured using luminescent plate reader after 15 min. Data was collected and processed using Gen5 software.

**Measurement of NAD:NADH ratio in chondrocytes**. Chondrocytes were treated with IL-1β (10 ng/mL) for 36 h in the presence or absence of IKK2 inhibitor (10 μM) or FX11 (40 μM). NAD and NADH levels were measured using NAD/NADH quantitation colorimetric kit (K337, Biovision, Milpitas, CA, USA). NAD:NADH ratio was measured for each sample. Assay was performed on 96-well plate and values measured using colorimetric plate reader.

**Measurement of GSH:GSSH ratio in chondrocytes**. Murine chondrocytes were treated with IL-1β (10 ng/mL) for 24 h in the presence of absence of FX11 (40 μM) or 6-ANA (50 mM). GSSG and total glutathione levels were measured using Glutathione Fluorometric Assay Kit (K264, Biovision, Milpitas, CA, USA). Fluorescence was measured in fluorescence plate reader in 96-well format. GSH levels were determined by subtracting GSSG levels from Total glutathione levels.

**Mitochondrial biogenesis assay**. Murine chondrocytes were treated with IL-1β (10 ng/mL) for 24 h in the presence of absence of SC-514 (10 μM). Mitochondrial biogenesis was measured using colorimetric Mitobiogenesis In-Cell ELISA kit (Ab110217, Abcam, Cambridge, MA, USA). Colorimetric assay was performed in 96-well plate format.

**Measurement of G6PD activity in chondrocytes**. Primary chondrocytes were plated in 6-well plates and treated with IL-1β for 24 h. Lysates were collected and assay was performed in 96-well plate following protocol of G6PD activity colorimetric assay kit (Cat# K751, Biovision, G6PD assay kit). Kinetic absorbance measurements were performed for 1 h using microplate reader and data was analyzed using Gen5 sotware. G6PD activity was derived by choosing activity between two timepoints in the linear region of curves.

**Measurement of NADPH levels**. Primary chondrocytes were plated in 10 cm dishes and treated with IL-1β for 24 h. Cell lysates were processed and NADPH levels were measured using fluorometric NADP/NADPH assay kit (Cat# ab17672, Abcam). Fluorescence was measured using microplate reader in black walled, clear bottom plates (Fisher). Data was collected and processed using Gen5 software.

**Quantification of ROS species**. Primary chondrocytes were treated for 24 h in DMEM media. Cells were washed two times with phenol red-free PBS, followed by incubation with 10 μM DCFDA (Cat#D6883, Sigma) or 5 μM DHE (Cat# D7008, Sigma) in PBS for 30 min, followed by two more washes with PBS. Cells were incubated in 37 °C incubator for 1 h in PBS, followed by fluorescence measurement using microplate reader using Ex/Em 495/525 for DCFDA and Ex/Em 480/576 for DHE.

**RNA sequencing and analysis**. Primary chondrocytes were cultured in the presence or absence of IL-1β (10 ng/mL) for 24 h. RNA was collected as stated above. Samples were prepared according to library kit manufacturer's protocol, indexed, pooled, and sequenced on an Illumina HiSeq. Basecalls and demultiplexing were performed with Illumina's bcl2fastq software and a custom python demultiplexing program with a maximum of one mismatch in the indexing read. RNA-seq reads were then aligned to the Ensembl release 76 primary assembly with STAR version 2.5.1a. Gene counts were derived from the number of uniquely aligned unambiguous reads by Subread:featureCount version 1.4.6-p5. Isoform expression of known Ensembl transcripts were estimated with Salmon version 0.8.2. Sequencing performance was assessed for the total number of aligned reads, total number of uniquely aligned reads, and features detected. The ribosomal fraction, known junction saturation, and read distribution over known gene models were quantified with RSeQC version 2.6.2.

All gene counts were then imported into the R/Bioconductor package EdgeR and trimmed mean of M-values (TMM) normalization size factors were calculated to adjust for samples for differences in library size. Ribosomal genes and genes not expressed in the smallest group size minus one samples greater than one count-per-million were excluded from further analysis. The TMM size factors and the matrix of counts were then imported into the R/Bioconductor package Limma. Weighted likelihoods based on the observed mean-variance relationship of every gene and sample were then calculated for all samples with the

voomWithQualityWeights. The performance of all genes was assessed with plots of the residual standard deviation of every gene to their average log-count with a robustly fitted trend line of the residuals. Differential expression analysis was then performed to analyze for differences between conditions and the results were filtered for only those genes with Benjamini–Hochberg false-discovery rate adjusted $p$-values ≤ 0.05.

For each contrast extracted with Limma, global perturbations in known Gene Ontology (GO) terms, MSigDb, and KEGG pathways were detected using the R/Bioconductor package GAGE[8] to test for changes in expression of the reported log2 fold-changes reported by Limma in each term versus the background log2 fold-changes of all genes found outside the respective term. The R/Bioconductor package heatmap3 was used to display heatmaps across groups of samples for each GO or MSigDb term with a Benjamini–Hochberg false-discovery rate adjusted $p$-value ≤ 0.05. Perturbed KEGG pathways where the observed log2 fold-changes of genes within the term were significantly perturbed in a single-direction versus background or in any direction compared to other genes within a given term with $p$-values ≤ 0.05 were rendered as annotated KEGG graphs with the R/Bioconductor package Pathview.

**Statistical analysis.** All experiments were repeated at least three times with similar results, unless otherwise stated in the figure legend. All images of immunoblots are representative of one of three experiments. Statistical analyses were performed using appropriate statistical test using GraphPad Prism and graphs were generated using Prism. Data from multiple experiments was displayed as mean ± S.E.M. DCFDA, viability, and seahorse experiment data is displayed as mean ± S.D. representative of one of multiple experiments. Multiple treatments were analyzed by one-way ANOVA followed by Tukey's test multiple comparisons test. Student's $t$-test was used for comparing two groups. Data from experiments was not combined if values between experiments showed large variations, but the trends remained the same between the tested conditions. Instead data is shown with technical replicates from one experiment, but statistics were not performed on technical replicates and no error bars are displayed for experiments with $n < 3$ replicates. Gene expression results from human OA cartilage was compared by paired student's $t$-test between patient's lateral and medial cartilage. Animal experiments were performed with age-matched, sex matched mice. $p$-values are indicated where applicable.

**Reporting summary.** Further information on research design is available in the Nature Research Reporting Summary linked to this article.

## Data availability
RNA-sequencing data have been deposited in Gene Expression Omnibus (GEO) under the primary accession code GSE148159. Other data that support the findings of this study are available from the corresponding author upon reasonable request. Source data are provided with this paper.

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

## Acknowledgements
This work was supported by NIH/NIAMS R01-AR049192, R01-AR054326, R01-AR072623 (to Y.A.), Biomedical grant from Shriners Hospital for Children #85160 (Y.A.), P30 AR057235 NIH Core Center for Musculoskeletal Biology and Medicine (to Y.A.), R01-AR075860 (to J.S.), and NIH/NIAMS R01-AR064755 and R01-AR068972 (to G.M.). We thank Dr. Mitchell C. Coleman from University of Iowa Health Care for insightful discussions.

## Author contributions
M.A., G.S., and J.S. participated in experimental design and troubleshooting. M.A., G.S., J.S., J.Y., X.D., and J.O. performed the experiments. M.F.R. provided human chondrocytes and analyzed related data. M.A. and G.S. prepared figures, participated in manuscript writing and revisions. T.M., contributed IκB-ζ floxed mice. G.M., K.K., and R.J.O. participated in experimental design, provided reagents and participated in manuscript writing. Y.A. planned the general outline of the project, guided the experimental approach throughout the duration of the project, and finalized the manuscript.

## Competing interests
The authors declare no competing interests.
