## [Peer Review File · Nature Communications]

REVIEWER COMMENTS

Reviewer #2 (Remarks to the Author):

Figure S3: The data regarding the effect of FX11 on lactate production remain troubling:

- a. The data in panel S3D show that FX11 inhibits IL-1b induced increases in ECAR. However, in S3H, FX-11 has no effect at all on lactate production by IL-1b activated cells. This result is confusing and not easily explained by evoking a role of LDHB, since even if LDHB is active in the FX11 cultures (and absence of LDHA function), we would expect to see ECAR remain high in the IL-1b plus FX11 condition, as is the case for lactate. I wonder if the answer to this question rests in the ways in which the experiments for S3D and S3H were performed – they were not done in the same way, and for S3D for example, measurements were made after the addition of oligomycin. Why was this? What do the data look like for lactate in the presence of oligo, and for ECAR in the absence?
- b. Using the MTT assay to measure cell viability in a system in which metabolic enzymes are being inhibited is probably not a great idea.

Figure. 6: The authors argue that my concerns about ROS regulating I κ B-z stability reflect a misreading of the results. They say that H₂O₂ increases I κ B-z in the presence of FX11 compared to FX11 alone. However, I cannot see this condition in Fig. 6G. I see only that compared to IL-1b alone, there is less I κ B-z in the H₂O₂ plus IL-1b condition. Compared to IL-1b alone, here is less I κ B-z in the FX11 plus IL-1b condition, but I do not see a condition where H₂O₂ and FX11 are added together.

The NAMPT data. The authors argument to keep these in the paper because they are “interesting in a roundabout way” is weak, and I recommend removing this section. Perhaps rework for a future publication.

Reviewer #4 (Remarks to the Author):

This is a revised manuscript originally reviewed by Nature Metabolism. The authors have done a very nice job responding to the reviewer comments and making appropriate revisions. There are several weaknesses in the experiments examining the role of ROS. These should at least be discussed as limitations in the discussion section.

1. The DCFDA probe used to measure ROS is quite non-specific and under certain conditions generates ROS and autofluorescence by itself and so is not longer recommended by those in the redox field to be used as a measure of ROS. This weakness is tempered by the use of a second probe (DHE). Redox biosensors are now the method of choice for ROS measures.
2. DPI was used as a Nox inhibitor but it has well documented issues with specificity because it inhibits multiple flavoenzymes. More specific NOX inhibitors are available such as VAS2870 or gp91ds-tat.
3. NAC and DMSO were used as anti-oxidants. These are also very non-specific reagents that have effects beyond simple ROS inhibition.

Response to Reviewer's comments:

We would like to thank the reviewers again for their time in reviewing our submission to Nature Communications. We feel that your comments have helped us improve the quality of our manuscript and its scientific merit. We have made changes to the manuscript taking into account the comments made by the reviewers as well as comments made regarding the editorial policy. We have addressed each of the points below:

Reviewer #2 (Remarks to the Author):

Critique 1:

Figure S3: The data regarding the effect of FX11 on lactate production remain troubling:

a. The data in panel S3D show that FX11 inhibits IL-1 β induced increases in ECAR. However, in S3H, FX-11 has no effect at all on lactate production by IL-1 β activated cells. This result is confusing and not easily explained by evoking a role of LDHB, since even if LDHB is active in the FX11 cultures (and absence of LDHA function), we would expect to see ECAR remain high in the IL-1 β plus FX11 condition, as is the case for lactate. I wonder if the answer to this question rests in the ways in which the experiments for S3D and S3H were performed – they were not done in the same way, and for S3D for example, measurements were made after the addition of oligomycin. Why was this? What do the data look like for lactate in the presence of oligo, and for ECAR in the absence?

b. Using the MTT assay to measure cell viability in a system in which metabolic enzymes are being inhibited is probably not a great idea.

Response 1a: The reviewer brings up an excellent point for which we are grateful. Indeed, the difference was in the conditions under which these experiments were performed. We observed that when we compared the Seahorse data prior to oligomycin administration that there is no difference in ECAR with FX11 treatment. This aligns well with the lactic acid data with FX11 treatment (Supplemental Figure 3G), as the reviewer concluded. However, it is true that upon oligomycin treatment, the FX11 treated cells start to display a decrease in ECAR. This is an interesting finding and something we want to pursue further in the future. In the manuscript, we replaced the ECAR data with that measured prior to oligomycin administration (Supplemental Figure 3D).

Response 1b: This is definitely an important point and one that is well taken. For future work, we will move away from the use of MTT and switch to a different measure of cellular viability. However, we are confident that FX11 treatment in our cells does not cause cell death. We have observed this through mRNA and protein content measurements, cell counts, and visual observation, as we have now stated in the discussion.

Critique 2:

Figure. 6: The authors argue that my concerns about ROS regulating I κ B-z stability reflect a misreading of the results. They say that H₂O₂ increases I κ B-z in the presence of FX11 compared to

FX11 alone. However, I cannot see this condition in Fig. 6G. I see only that compared to IL-1b alone, there is less I κ B-z in the H₂O₂ plus IL-1b condition. Compared to IL-1b alone, here is less I κ B-z in the FX11 plus IL-1b condition, but I do not see a condition where H₂O₂ and FX11 are added together.

Response 2: This was a mistake on our part. We realize now that the figure was mislabeled. We apologize for that and appreciate your keen observation. The labels for Figure 6G have now been corrected to reflect the accurate conditions which include an FX11 + H₂O₂ condition, consistent with the original description of the data. In addition, we have included gene expression data in Supplemental Figure 6J displaying that H₂O₂ can reverse some of FX11's inhibitory effect on IL-6 expression.

Critique 3:

The NAMPT data. The authors argument to keep these in the paper because they are "interesting in a roundabout way" is weak, and I recommend removing this section. Perhaps rework for a future publication.

Response 3: We understand and agree that the NAMPT data is distracting and unclear in the context of the overall story we are proposing, especially given the intricate role of NAD in all aspects of metabolism. Thus, we have removed all the data regarding the role of NAMPT from the paper and will develop that story further to gain a better mechanistic understanding of the role of NAD in the inflammatory response.

Reviewer #4 (Remarks to the Author):

This is a revised manuscript originally reviewed by Nature Metabolism. The authors have done a very nice job responding to the reviewer comments and making appropriate revisions. There are several weaknesses in the experiments examining the role of ROS. These should at least be discussed as limitations in the discussion section.

Critique 1. The DCFDA probe used to measure ROS is quite non-specific and under certain conditions generates ROS and autofluorescence by itself and so is not longer recommended by those in the redox field to be used as a measure of ROS. This weakness is tempered by the use of a second probe (DHE). Redox biosensors are now the method of choice for ROS measures.

Response 1: This is a very good point. We understand the drawbacks of using fluorescent probes and want to utilize more specific ROS readouts such as Redox sensors moving forward. We bring this up in the discussion as a limitation of this study. The use of redox biosensors could allow for a much better understanding of the mechanism at hand.

Critique 2. DPI was used as a Nox inhibitor but it has well documented issues with specificity because it inhibits multiple flavoenzymes. More specific NOX inhibitors are available such as VAS2870 or gp91ds-tat.

Response 2: Knowing that DPI can have off-target effects on other flavoenzymes, especially those found in the mitochondria, we utilized Vas2870 and observed similar inhibitory effects on ROS as measured by DCFDA and the inflammatory response. We now included this data in Supplemental Figure 4H-I and supplemental 6G-H. We feel that this supports the claim that NOX enzymes are important for mediating the chondrocyte inflammatory response.

Critique 3. NAC and DMSO were used as anti-oxidants. These are also very non-specific reagents that have effects beyond simple ROS inhibition.

Response 3: This is true that these compounds can act as anti-oxidants but can also have broad effects, especially DMSO. We feel that the use of two "antioxidants" along with the corresponding hydrogen peroxide data support the idea of I κ B- ζ being redox sensitive. However, we understand that these are not precise tools and as such, we have included in the discussion section that there can be off-target effects of compounds such as NAC, making this a limiting factor in the study. In the future, we want to use more targeted compounds or genetic manipulation to answer questions about I κ B- ζ stability.

REVIEWERS' COMMENTS:

Reviewer #2 (Remarks to the Author):

The paper has been adequately revised and is now acceptable for publication in my opinion.